# Effects of Liquid Phase Nano Titanium Dioxide (TiO_2_) on Seed Germination and Seedling Growth of Camphor Tree

**DOI:** 10.3390/nano12071047

**Published:** 2022-03-23

**Authors:** You Zhou, Lei Zhang, Yaodi Liu, Jiyun She

**Affiliations:** 1College of Forestry, Central South University of Forestry and Technology, Changsha 410004, China; zhouyouhuhst@126.com; 2School of Marxism, Hunan University of Humanities, Science and Technology, Loudi 417000, China; 3Ulink College of Shanghai, Shanghai 201615, China; lei.zhang01@ulink.com; 4College of Science, Tibet University, Lhasa 850000, China; Liuyd0222@163.com

**Keywords:** camphor tree, nano titanium dioxide, seed germination, seedling growth, concentration

## Abstract

It is of great significance to popularize and apply nanotechnology in forest plantations for the high-quality development of such areas. Camphor trees have good ecological and environmental benefits and are economic, which makes them worthy of widespread popularization and promotion. In this paper, we successfully synthesized bulk and rod-like TiO_2_ powder and used it to study the influence of camphor seed germination and seedling growth. The germination rate, germination potential, germination index activity index of camphorwood seed during germination were measured by TiO_2_ solution with different morphology. Meanwhile, the fresh weight, root length and seedling height of seedlings, as well as the activities of CAT, SOD and POD and MDA content in the seedlings were measured in detail. The difference in the promoting effect between bulk and rod TiO_2_ powder was compared. The possible reasons are also explained. The results showed that bulk and rod-like TiO_2_ solution improved the activities of SOD, POD and CAT, and increased the resilience of camphor seedlings. Moreover, the rod-like TiO_2_ solution has a stronger osmotic effect on seed, and has a better effect on promoting seed germination and seedling growth. The study on the influence of nano-TiO_2_ concentration also further showed that the treatment of nano-TiO_2_ solution with appropriate concentration could effectively promote seed germination and seedling growth, and enhance its adoptability to adversity; but excessive concentration will bring some side effects, which was not conducive to seed germination and seedling growth. In general, the results of this study provide a theoretical basis and technical guidance for the practical application of nanotechnology in camphor seedling and afforestation production.

## 1. Introduction

Camphor trees are evergreen in four seasons, with a strong vitality, low soil requirement and long survival period [1], and they can be used as street trees or landscape trees. They have a good ability to protect the environment by absorbing fumes from smoking and dust, helping to form headwaters, causing sand fixation, and making the environment more picturesque. In addition, the volatile oil made from camphor trees, which as a special aroma, has many benefits, such as temperature resistance, corrosion resistance and insect elimination, and can be used to make mothballs. The trunk and fruit can be used for medicine, and the medicine made from roots is called Tu Chen Xiang, which has the effects of promoting qi, promoting blood to reduce stasi, and removing arthralgia caused by wind, moisture or cold temperatures [2]. The latest research shows that camphor tree oil also has the function of reducing visceral fat and improving blood lipids [3]. Based on the above characteristics, the camphor tree was selected as the city tree of Loudi City, Hunan Province in 2004.

At present, camphor trees have the problems of low seed yield, low germination rate and low seedling rate in artificial cultivation, which have become major issues to solve in current large-scale promotion and planting work [4]. Improving the germination rate and seedling growth activity is the primary technical problem regarding the artificial cultivation of the camphor tree. Therefore, it is of great practical significance to seek effective seed-soaking methods to improve seed germination and seedling growth. 

In recent years, nano-TiO_2_ has been broadly applied in the field of biology due to its special spatial conformation, micro-electromagnetic properties, certain hormone effects and antibacterial effects [5,6,7]. Some researchers found that the appropriate concentration of TiO_2_ solution could improve the effects of superoxide dismutase (SOD), peroxidase (POD) and catalase (CAT) at the germination stage, thereby accelerating seed germination, improving the survival rate of seedlings and improving the stress resistance of plants, which is equivalent to improving seed vigor, promoting seed germination rate and seedling growth [8,9,10]. However, most of the existing studies focus on the regulation of nano-TiO_2_ in the growth of various microorganisms and lower plants, while there are few studies on the effect of nano-TiO_2_ on the growth of higher plants. The difference in the promoting effect of nano-TiO_2_ with a different structure and morphology on camphor seed germination is rarely reported. 

Based on the above context, in this paper, bulk and rod-like nano-TiO_2_ powders were synthesized and configured into a certain concentration of solution for soaking camphor seed. Taking the germination rate, germination potential, germination index and vigor index of camphor seed as indexes, combined with the fresh weight, root length and seedling height of camphor seedlings; the activities of catalase (CAT), superoxide dismutase (SOD), peroxidase (POD); and the malondialdehyde (MDA) content in seedlings, the effects of TiO_2_ solutions with different structures and concentrations on the germination of camphor seed were investigated. The objective of this study is to develop the application of nano-TiO_2_ in the field of seedling cultivation, which is of great significance to the high-quality development of camphor artificial planting technology.

## 2. Materials and Methods

### 2.1. Synthesis of Bulk and Rod-like TiO_2_ Nanoparticles

All the chemicals were used as received.

There were some changes made to the synthesis of spherical nano-TiO_2_ according to the literature [11]. In a typical synthesis, 10 mL tetra-isopropyl orthotitanate was dissolved in 90 mL isopropoxide (TTIP) to form homogeneous solution. Then, 5 mL distilled water was added to the solution in terms of a molar ratio of Ti:H_2_O = 1:4. Nitric acid was used to adjust the pH to restrain the hydrolysis process of the solution. The solution was vigorously stirred for 50 min. After aging for 36 h, the sols were transformed into gels. The obtained gels were dried under 100 °C for 4 h to evaporate water and organic material to the maximum extent. Then, the dry gel was sintered at 400 °C for 2 h. The dried powder was ground by agate mortar, using pestle to remove agglomerates to obtain spherical TiO_2_ nanoparticles, and marked as TiO_2_-bulk.

There were some changes made to the synthesis of rod-like TiO_2_ according to the literature [12]. Firstly, 1:6 molar ratio of bulk TiO_2_ (<5 µm, 0.05 moL/L) to sodium hydroxide (NaOH, 1.0 moL/L) solution was dissolved in 300 mL DI water, with a mild stirring at room temperature; and in a 500 mL glass beaker with vigorous stirring for 90 min at 100 °C. After 120 min, the precipitate was collected and centrifuged at 3000 r/min for 5 min. This process was repeated 3 times using ethanol. The remainder was dried in a furnace at 80 °C for 12 h. After drying the substance, it was hand ground in agate mortar with a pestle to turn it into powder form. The powder was annealed at 600 °C for 2 h and marked as TiO_2_-rod.

The structures and morphologies of TiO_2_-bulk and TiO_2_-rod were characterized by XRD and SEM.

### 2.2. Test on Camphor Seed Germination

The seed of Chinese fir were cleaned with pure water 5 times, disinfected with 0.1% (mass fraction) potassium permanganate solution for 3 h; and then cleaned with pure water to clear and remove, while standing for 3 h, the floating inferior seed. The treated full seed were immersed in nano-TiO_2_ solution (TiO_2_-bulk and TiO_2_-rod) with diverse concentration (50, 100, 200 and 500 mg/L) at 45 °C for 24 h, pure water treatment was used as the control group. Thirty seeds with full particles and uniform sizes were selected from each group. After immersion, those 30 seeds were cultured in an artificial climate chamber in a disposable Petri dish, which had a diameter of 9 cm and two layers of pure water infiltration filter paper. Culture conditions: at 25 °C, illumination 12 h, darkness 12 h, humidity 75%, illumination 4200 lx. 

### 2.3. Determination of Seed Germination Index and Morphological Index

The number of seed germinations was observed every 12 h after the experiment. Seed germination was marked by a 2 mm radicle breaking through of the seed coat, and was discarded following 7 consecutive days without new seed germination. The germination rate, germination potential, germination index and vigor index of seed were calculated according to standard [13]. Under the same conditions for 15 days, the root length, seedling height and fresh weight of all camphor seedlings treated with different concentrations of nano-TiO_2_ solution were measured. Formulas of germination rate (G), germination potential (GP), germination index (GI) and vigor index (VI) are as follows: G = (amount of seeds germinated/amount of seed samples) × 100%
GP = (amount of seeds reaching germination peak/amount of seed samples) × 100%
GI = ∑ (Gt/Dt)
VI = GI × S

In the formulas, Gt is the amount of germination in time t, Dt is the corresponding germination days, and S is the length of seedling root (cm).

### 2.4. Determination of Physiological Indexes of Seedling

Preparation of enzyme solution: 0.4 g germinated camphor seedlings were washed and dried, and put into the ice bath bowl. Subsequently, 4 mL phosphate-buffered solution (0.005 moL/L PBS pH 7.8) was added for rapid grinding to homogenize, and then 6 mL phosphate-buffered solution was added for flushing to the centrifuge tube. At 4 °C, 10,000 r/min—1 for 30 min, the supernatant acted as the enzyme solution to be tested and was placed in a 4 °C refrigerator. Superoxide dismutase (SOD), peroxidase (POD), catalase (CAT) and malondialdehyde (MDA) were measured according to literature [14] and the process explained below:

For SOD activity measurement: fresh Camphor tree seedlings (0.60 g) were ground thoroughly with a cold mortar and pestle in potassium phosphate buffer (pH 7.0, 50 mM) with 0.1 mM EDTA. The homogenate was centrifuged at 20,000 rmp for 40 min at 4 °C. The supernatant was crude enzyme extraction. Activity of SOD was measured by the photochemical method with nitro-blue tetrazolium. One unit of SOD activity was defined as the amount of enzyme required to give 50% inhibition of the rate of nitro-blue tetrazolium.

POD activity measurement: fresh Camphor tree seedlings (1.0 g) were grinded in an ice bath in 5 mL borate buffer (pH 8.7, 50 mM) containing sodium hydrogen sulfite (5 mM) and Polyvinyl Pyrrolidone (0.1 g). Centrifuging the homogenate at 10,000 rmp at 4 °C for 40 min to obtain the enzyme extraction. A substrate mixture, which contains acetate buffer (0.1 mM, pH 5.4), ortho-dianisidine (0.25% in ethyl alcohol) and 0.75% H_2_O_2_ (0.1 mM) was added to the enzyme extract (0.1 mL). POD activity was determined based on the change in absorbance of the brown guaiacol at 460 nm. AT activity was determined using UV-VIS spectra by measuring the decrease in absorbance at 240 nm for 1 min following the decomposition of H_2_O_2_. The reaction mixture contained 50 mM phosphate buffer (pH 7.0), 15 mM H_2_O_2_ and 0.1 mL enzyme extract. CAT activity was calculated from the extinction coefficient (40 mM^−1^ cm^−1^) of H_2_O_2_.

MDA contents measurement: fresh Camphor tree seedling sample (1.0 g) was added to a total of 15 mL aqueous solution, which consisted of 7.5% trichloroacetic acid (*w*/*v*) with 0.1% (*w*/*v*) of ethylenediaminetetraacetic acid and 0.1% (*w*/*v*) of propyl gallate. The mixture was homogenized with an homogenizer (Scientz-150, Ningbo, China) for 1 min at 20,000 rpm, and the volume was adjusted to 30 mL with the addition of trichloroacetic acid. The homogenate was filtered through 150 mm filter paper, and a specific volume reacted with the thiobarbituric acid reagent. HPLC was used for MDA quantification. Briefly, a total of 1 mL of extract and 3 mL of thiobarbituric acid reagent (40 mM dissolved in 2 M acetate buffer at pH 2.0) were mixed in a test tube and heated in a boiling water bath for 35 min. The reaction mixture was chilled prior to the addition of 1 mL of methanol, and 20 μL of the sample were injected into a Varian C18 HPLC column (5 μm, 150 × 4.6 mm) and held at 30 °C. The mobile phase consisting of 50 mM KH_2_PO_4_ buffer solution, methanol, and acetonitrile (72:17:11, *v*/*v*/*v*, pH 5.3) was pumped isocratically at 1 mL min^−1^.

## 3. Results

### 3.1. XRD and SEM Characterization of Nano-TiO_2_ with Different Morphologies

XRD and SEM characterization of bulk-TiO_2_ and rod-like TiO_2_ powders, as shown in Figure 1 and Figure 2.

Figure 1 shows the SEM image of obtained TiO_2_ nanoparticles of a spherical and rod morphology. The TiO_2_-bulk sphere has a single shape and uniform dispersion. Additionally, the clear TiO_2_-bulk nanostructures are seen to have a grain size of ~200 nm. Moreover, it is clear show that TiO_2_-bulk spheres consist of a number of crystallites, which are seen by the TEM image. Thus the SEM characterization indicates the successful synthesis of TiO_2_ with different morphologies [15].

The XRD patterns of obtained TiO_2_ nanoparticles of spherical and rod morphology are shown in Figure 2a,b, respectively. The synthesized nanoTiO_2_ both showed a crystalline nature with 2θ peaks at 2θ = 25.25° (101), 2θ = 37.8° (004), 2θ = 47.9° (200), 2θ = 53.59° (105) and 2θ = 62.36° (204). All the peaks in the XRD patterns could be indexed as anatase phases of TiO_2_ and the diffraction data were in good agreement with JCPDS files # 21-1272 [15]. So, the SEM and XRD characterizations illustrate the successful synthesis of desired TiO_2_-bulk and TiO_2_-rod nanomaterials.

### 3.2. Germination Rate, Germination Potential, Germination Index and Vigor Index of Camphor Seed Treated with Nano-TiO_2_ Solutions with Different Morphologies

The germination rate, germination potential, germination index and vigor index of camphor seeds after soaking with 100 mg/L bulkTiO_2_ and rod TiO_2_ are shown in Figure 3.

Figure 3 shows that the indexes of camphor seed soaked in bulk TiO_2_ and rod-like TiO_2_ solutions increased to different degrees. The germination rate is increased from 42 ± 8.1% of the blank sample to 68 ± 12.4% and 89 ± 14.5%, respectively. The germination potential is increased from 15 ± 3.4 of the blank sample to 20 ± 4.1 and 28 ± 0.52, respectively. The germination index is increased from 2.0 ± 0.4 of the blank sample to 3.1 ± 0.5 and 4 ± 0.7, respectively. The vigor index is increased from 3.6 ± 0.7 of the blank sample to 13.02 ± 2.7 and 22 ± 4.1, respectively.

It has been reported that plant seed soaked in nano titanium dioxide solution can absorb nanoparticles to varying degrees. The absorbed nanoparticles can promote germination in seeds [16]. Specifically, there are two reasons that liquid-phase TiO_2_ solution promotes camphor tree seed germination. One reason is that the concentration of nano-TiO_2_ enhances the activity of water to help the seeds absorb water, activate seed cells and tissues, stimulate organism function, and promote the germination and growth of Chinese fir seed. On the other hand, the seed enzyme activity is stimulated, making the free water in the seed become bound water, thereby promoting its germination. The promoting effect of rod-like TiO_2_ solution on seed germination is clearer than that of bulk TiO_2_. The reason for this may be that the shape of rod-like TiO_2_ in the radial direction is easier to penetrate in the organizational structure of the seed, helping the seed to absorb more water and better promote seed germination [17,18,19].

### 3.3. Fresh Weight, Root Length, Seedling Height, CAT, SOD, POD Activity and MDA Content of Seedlings in Nano-TiO_2_ Solution with Different Morphologies

Fresh weight, root length, seedling height, CAT, SOD, POD activity and MDA content of germinated camphor seedlings soaked with 100 mg/L bulk TiO_2_ and rod TiO_2_ are shown in Table 1.

Table 1 shows that, compared with the blank sample, the fresh weight and root length of camphor seed germination seedlings after impregnation show a clear promotion, and the promotion effect of rod-like TiO_2_ solution is better than bulk TiO_2_ solution. However, the effect of different morphologies of TiO_2_ solutions on increasing seedling height is not clear. The contents of CTA, SOD and POD in the seedlings germinated after impregnation increase at different rates, while the content of MDA decreases.

Increasing the activities of CAT, SOD and POD can effectively eliminate excessive free radicals in plants, reduce membrane lipid peroxidation, and maintain the integrity of the structure and function of the membrane system [20,21]. In this study, the CAT, SOD and POD activities of camphor seed germination seedlings soaked in bulk TiO_2_ solution and rod TiO_2_ solution were significantly increased, because TiO_2_ can induce oxygen-free radicals. Oxygen-free radicals produced at suitable concentrations act as signals to induce the gene expression of the antioxidant enzyme system, thereby promoting the synthesis of several protective enzymes and improving the antioxidant capacity of wood seedlings [22,23]. MDA is one of the main products of lipid peroxidation in plant biofilm systems. It can cross-link with biological macromolecules, such as proteins and nucleic acids, changing the configuration of biological macromolecules and even making them lose their functions. Therefore, the amount of MDA can be used as an important indicator of the intensity of membrane lipid peroxidation and the damage degree of membrane system. In this study, different morphologies of TiO_2_ solution showed a good inhibitory effect on MDA in seedlings, which is another piece of evidence that it increases seedling growth.

### 3.4. Fresh Weight, Root Length and Seedling Height of Seedlings Treated with Different Concentrations of TiO_2_—Rod Solution

In order to further investigate the effect of the concentration of rod-like TiO_2_ in the impregnation solution, rod-like TiO_2_ with a good increasing effect on the germination and seedlings of cinnamomum camphor was taken as the object, and 0, 50, 100, 200 and 500 mg/L TiO_2_ solutions were selected for comparison. The fresh weight, root length and seedling height of camphor seedlings increased germination, as shown in Figure 4.

In Figure 4, compared with the blank sample, the fresh weight and root length of camphor seed germination seedlings soaked with different concentrations of rod-like TiO_2_ solution show a clear increase. However, when the solution concentration is greater than 100 mg/L, the seedling fresh weight stops increasing and has a promoting effect on root length. Different concentrations of rod-like TiO_2_ solution do not significantly promote seedling height.

As discussed in Section 3.3, different morphologies of TiO_2_ solution can promote the synthesis of several protective enzymes in seedlings and improve the antioxidant capacity of cinnamomum camphora seedlings. However, when the concentration of nanomaterials is too high, a large number of oxygen free radicals will be produced, destroying the cell structure and weakening these promoting effects [24]. Additionally, when the solution concentration is too high, camphor seeds will produce a cumulative effect in the process of seed soaking, resulting in a different sensitivity in the process of seedling growth and more complex results [7].

There are, of course, current records of the increasing effect of TiO_2_ solution with different morphologies on the seed germination and seedling growth of cinnamomum camphora, which are mostly based on the existing literature, and offer no more original evidence. In order to have a deeper understanding of the reasons behind this, the cross-disciplinary cooperation of scientists in the field of materials, forest experts and researchers other disciplines is necessary.

## 4. Conclusions

In this paper, bulk and rod-like TiO_2_ powders were successfully synthesized and their effects on the seed germination and seedling growth of cinnamomum camphora were studied. The germination rate, germination potential, germination index and vigor index of cinnamomum camphora seeds treated with different morphologies of TiO_2_ solution were measured in detail. At the same time, the fresh weight, root length, seedling height, CAT, SOD, POD activity and MDA content in seedlings were recorded in detail. The data showed that bulk and rod-like TiO_2_ solutions significantly improved the SOD, POD, CAT activity and stress resistance of camphor seedlings. In particular, rod-like TiO_2_ solutions have a stronger osmotic effect on seeds, which is more effective for promoting seed germination and seedling growth than bulk TiO_2_ solutions. However, the mechanism requires the interdisciplinary cooperation and a further in-depth study. Overall, the results of this study provide a theoretical basis and technical guidance for nano-technology (especially nano-TiO_2_) in camphor seedlings and afforestation production, as well as better technical support for the promotion of camphor tree planting and high-value utilization.

## Figures and Tables

**Figure 1 nanomaterials-12-01047-f001:**
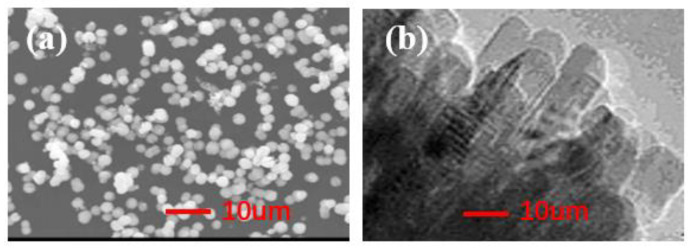
SEM characterization of TiO_2_-bulk (**a**) and TiO_2_-rod (**b**).

**Figure 2 nanomaterials-12-01047-f002:**
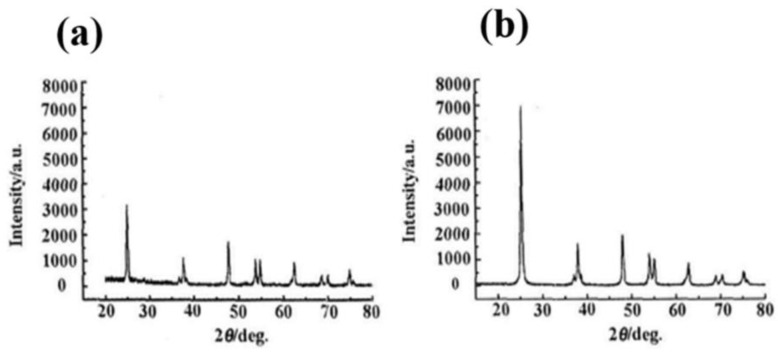
The XRD patterns of TiO_2_-bulk (**a**) and TiO_2_-rod (**b**).

**Figure 3 nanomaterials-12-01047-f003:**
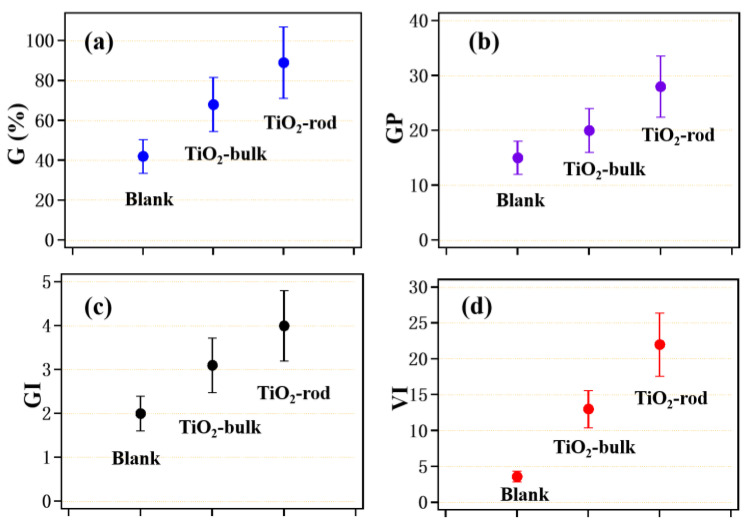
Germination rate (**a**); germination potential (**b**); germination index (**c**) and vigor index (**d**) of camphor seed soaked with liquid bulkTiO_2_ and rod TiO_2_ (concentration of 100 mg/L).

**Figure 4 nanomaterials-12-01047-f004:**
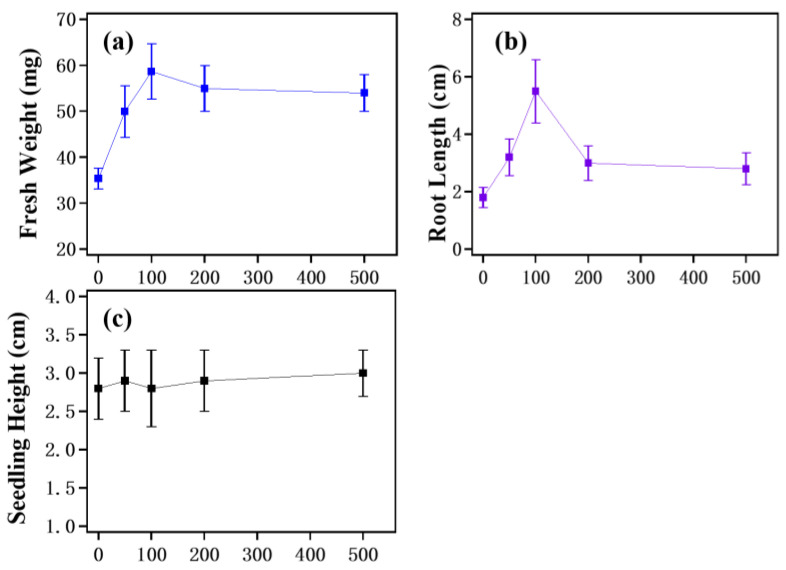
Fresh weight (**a**), root length (**b**) and seedling height (**c**) of camphor seeds germinated after soaking with different concentrations of rod-like TiO_2_.

**Table 1 nanomaterials-12-01047-t001:** Fresh weight, root length, seedling height, CAT, SOD, POD activity and MDA content of germinated camphor seedlings soaked with bulk TiO_2_ and rod TiO_2_.

Solution	Fresh Weight	Root Length	Seedling Height	CTA	SOD	POD	MDA
	mg	cm	cm	U/(g·min)	U/g	U/(g·min)	umoL/g
Blank sample	35.4 ± 2.3	1.8 ± 0.3	2.8 ± 0.4	5.2 ± 1.0	78.6 ± 12.2	48.6 ± 6.1	4.3 ± 0.8
TiO_2_-bulk	48.1 ± 4.2	4.2 ± 0.7	2.9 ± 0.3	11.3 ± 2.1	300.0 ± 42.5	69.7 ± 9.5	2.5 ± 0.4
TiO_2_-rod	58.7 ± 6.0	5.5 ± 0.9	2.8 ± 0.5	14.7 ± 2.4	393 ± 40.1	88.5 ± 11.8	2.1 ± 0.4

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
