# Peer review of "Effects of Liquid Phase Nano Titanium Dioxide (TiO_2_) on Seed Germination and Seedling Growth of Camphor Tree"

_nanomaterials, 2022, doi:10.3390/nano12071047_

Round 1
Reviewer 1 Report
According to manuscript Nanomaterials -1634147, entitled „Effects of Liquid phase nano titanium Dioxide (TiO2) on seed germination and seedling growth of Camphor tree“ by authors You Zhou and Jiyun She.
This manuscripts presents very interesting research on the impact of nano TiO2 powders on the seed germination and seedling of Camphor tree. The samples are nanostructures with sperical and rod-like TiO2. The XRD and morphological studies are performed. The other part of manuscripts presents research on germination rate, germination potential, germination index of seeds treated with different morphologies of TiO2 and weight, root length, seedling height in seedlings. The application of nanostructured materials in forest plantation is attractive and usefull. The manuscript contains interesting scientific results.
The paper can be accepted after major corrections.
- Experimental : The authors states „Synthesis of spherical nano-TiO2: there are a little bit changes according to the liter-72 ature[11].“ It is preferable to decribe your technological synthesis and then to state that it is similar to other methods in literature.
- Experimental: For the seed germination study the authors decsribe that they are immersed in „nano-TiO2 solution with different morphology“. It is very unclear what this solution is. Is it the precursor solution for synthesis of the TiO2 nanostructures? It must be clearly stated.
- More explanation for XRD results: TiO2 crystal phase and crystallite sizes are good to be given.
- SEM and XRD patterns are good to be two separate Figures
- Figure tables What is „liquid spherical TiO2 and rod TiO2“?
- As the Figure 2 caption is stated „Germination rate (a) ; germination potential (b) ; germination index (c) and vigor index (d) 140 of camphor seed soaked with liquid spherical TiO2 and rod TiO2 (concentration of 100 mg / L). But in the fugures it is given only TiO2 rod.
- The studies given in the second part of the manuscript need to be more clearly described. The experimental conditions, the measuring techniques and the parameters.
Author Response
Dear Editor and Reviewer1:
Thank you for your letter and for the reviewers’ comments concerning our manuscript for publication in Nanomaterials. These comments are all valuable and very helpful for revising and improving our paper, as well as the important guiding significance to our researches. We have studied comments carefully and have made corrections point-to-point which we hope meet with approval and are marked with yellow, green and blue (blue indicates language revision). The responds to the reviewers’ comments are as follows:
Reviewer #1: This manuscripts presents very interesting research on the impact of nano TiO2 powders on the seed germination and seedling of Camphor tree. The samples are nanostructures with sperical and rod-like TiO2. The XRD and morphological studies are performed. The other part of manuscripts presents research on germination rate, germination potential, germination index of seeds treated with different morphologies of TiO2 and weight, root length, seedling height in seedlings. The application of nanostructured materials in forest plantation is attractive and usefull. The manuscript contains interesting scientific results. Author reply: We thank the reviewers for the time and effort that you have put into reviewing the previous version of the manuscript. Your suggestions have enabled us to improve our work. Thank you so much for your positive evaluation. The point-to-point corrections are marked with yellow and blue (blue indicates language revision). 1. Experimental: The authors states „Synthesis of spherical nano-TiO2: there are a little bit changes according to the literature [11].“ It is preferable to describe your technological synthesis and then to state that it is similar to other methods in literature.Author reply: Thank you so much for your useful comments. The study mainly focuses on the exploration of nanotechnology in forest plantation for high-quality development of plantation. The descriptions of bulk-TiO2 and rod-TiO2 are insufficient. According your advice, some more detailed technological synthesis description is added in the revised manuscript and below (marked with yellow). Thanks again for your advices!All the chemicals were used as received.Synthesis of spherical nano-TiO2: there are a little bit changes according to the literature[11]. In a typical synthesis, 10 mL tetra-isopropyl orthotitanate was dissolved in 90 mL isopropoxide (TTIP)to form homogeneous solution. 5 mL distilled water was added to the solution in terms of a molar ratio of Ti: H2O=1:4. Nitric acid was used to adjust the pH to restrain the hydrolysis process of the solution. The solution was vigorously stirred for 50 min. After aging for 36 hrs, the sols were transformed into gels. The obtained gels were dried under 100oC for 4 hours to evaporate water and organic material to the maximum extent. Then the dry gel was sintered at 400oC for 2 hrs. The dried powder was ground by agate mortar using pestle to remove agglomerates to obtain spherical TiO2 nanoparticles and marked as TiO2-bulk. 2. Experimental: For the seed germination study the authors describe that they are immersed in nano-TiO2 solution with different morphology“. It is very unclear what this solution is. Is it the precursor solution for synthesis of the TiO2 nanostructures? It must be clearly stated.Author reply: Thank you so much for your careful check. In our study, we first synthesis the spherical and rodlike TiO2 nanoparticles (marked as TiO2-bulk and TiO2-rod). The seeds were immersed in the TiO2 aqueous solution (TiO2-bulk or TiO2-rod) at 45 oC for 24hrs. After immersion, the seeds were cultured them in an artificial climate chamber. To clearly illustrate the “Test on camphor seed germination” process, some details descriptions have been added and marked with yellow. The modified description also is shown as below.The seed of Chinese fir were cleaned with pure water for 5 times, disinfected with 0.1% (mass fraction) potassium permanganate solution for 3 hrs; and then cleaned with pure water to clarify, standing for 3 hrs, removed the floating inferior seed. The treated full seed were immersed in nano-TiO2 solution (TiO2-bulk and TiO2-rod) with diverse concentration (50, 100, 200 and 500 mg/L) at 45°C for 24hrs, pure water treatment was used as the control group. 30 seed with full particles and uniform size were selected from each group. After immersion, those 30 seeds were cultured in an artificial climate chamber in a disposable petri dish, which with diameter of 9 cm and two layers of pure water infiltration filter paper. Culture conditions: at 25 °C, illumination 12hrs, darkness 12hrs, humidity 75%, illumination 4200lx. 3. More explanation for XRD results: TiO2 crystal phase and crystallite sizes are good to be given.Author reply: Thank you so much for the suggestions that improves our paper. Figure 1 has corrected to two separate Figures and the more explanations have been given in the revised manuscript and below (marked with yellow). Figure 1 SEM characterization of TiO2-bulk (a) and TiO2-rod (b).Figure 1 shows the SEM image of obtained TiO2 nanoparticles of spherical and rod morphology. The TiO2-bulk sphere has single shape and uniform dispersion. And the clear TiO2-bulk nanostructures can be seen having grain size of ~ 200 nm. Besides, it is clear show that TiO2-bulk consist of a number of crystallites which are seen by TEM image the synthesized TiO2 are spherical and rod-like, the size of nanoparticles is uniform, and the dispersion is good. XRD results are also the characteristic diffraction peaks of TiO2, Thus the SEM character-ization indicating indicates the successful synthesis of TiO2 with different morphologies [15].
Figure 2. The XRD patterns of TiO2-bulk (a) and TiO2-rod (b)
The XRD patterns of obtained TiO2 nanoparticles of spherical and rod morphology are shown in Figure 2(a) and (b) respectively. The synthesized nanoTiO2 both showed crystalline nature with 2θ peaks lying at 2θ=25.25o (101), 2θ=37.8o (004), 2θ=47.9o (200), 2θ=53.59o (105) and 2θ=62.36o (204). All the peaks in the XRD patterns can be indexed as anatase phases of TiO2 and the diffraction data were in good agreement with JCPDS files # 21-1272 [15]. So, the SEM and XRD characterizations illustrate the successful synthesis of desired TiO2-bulk and TiO2-rod nanomaterials. 4. SEM and XRD patterns are good to be two separate Figures
Author reply: Thanks for your advice, the SEM and XRD patterns have been separated into two Figures and can be seen in the revised manuscript (and question 3 above).
5. Figure tables What is „liquid spherical TiO2 and rod TiO2“?
Author reply: Thanks for your advice. In this study, we synthesis the TiO2 nanomaterials, which have two morphology (spherical and rod as shown in Figure 1). Then, the aqueous solution of TiO2 were used to treat the seeds that further study the influence of camphor seed germination and seedling growth. In order to avoid confusion, the descriptions of “spherical TiO2” have been modified to “TiO2-bulk” in the revised manuscript and marked with yellow.
6. As the Figure 2 caption is stated „Germination rate (a); germination potential (b) ; germination index (c) and vigor index (d) 140 of camphor seed soaked with liquid spherical TiO2 and rod TiO2 (concentration of 100 mg / L). But in the fugures it is given only TiO2 rod.
Author reply: Thanks for pointing this. We are sorry for that due to our negligence, there are some inconsistencies in the description of spherical TiO2 in the paper. As raised in the above comment, the descriptions of “spherical TiO2” have been modified to “TiO2-bulk”.
7. The studies given in the second part of the manuscript need to be more clearly described. The experimental conditions, the measuring techniques and the parameters.
Author reply: Thanks for your valuable advice. More detailed experimental conditions, the measuring techniques and the parameters are given in the revised manuscript. specially, Superoxide dismutase (SOD), peroxidase (POD), catalase (CAT) and malondialdehyde (MDA) measurement details are shown as below.
For SOD activity measurement: Fresh Camphor tree seedling (0.60 g) were ground thoroughly with a cold mortar and pestle in potassium phosphate buffer (pH 7.0, 50 mM) with 0.1 mM EDTA. The homogenate was centrifuged at 20000 rmp for 40 min at 4 oC. The supernatant was crude enzyme extraction. Activity of SOD was measured by the photochemical method with nitro-blue tetrazolium. One unit of SOD activity was defined as the amount of enzyme required to give 50 % inhibition of the rate of nitro-blue tetrazolium.POD activity measurement: Fresh Camphor tree seedling (1.0 g) were grinded in an ice bath in 5 mL borate buffer (pH 8.7, 50 mM) containing sodium hydrogen sulfite (5 mM) and Polyvinyl Pyrrolidone (0.1 g). Centrifuging the homogenate at 10 000 rmp at 4oC for 40 min to obtain the enzyme extraction. A substrate mixture, which contains acetate buffer (0.1 mM, pH 5.4), ortho-dianisidine (0.25 % in ethyl alcohol) and 0.75 % H2O2 (0.1 mM) was added to the enzyme extract (0.1 mL). POD activity was determined based on the change in absorbance of the brown guaiacol at 460 nm.CAT activity was determined by UV-VIS spectra. By measuring the decrease in absorbance at 240 nm for 1 min following the decomposition of H2O2. The reaction mixture contained 50 mM phosphate buffer (pH 7.0), 15 mM H2O2 and 0.1 mL enzyme extract. CAT activity was calculated from the extinction coefficient (40 mM-1 cm-1) for H2O2.MDA contents measurement: Fresh Camphor tree seedling sample (1.0 g) was added into a total of 15 mL aqueous, which is of 7.5% Trichloroacetic acid (w/v) with 0.1% (w/v) of ethylenediaminetetraacetic acid and 0.1% (w/v) of propyl gallate. The mixture was homogenized with an homegenizer (Scientz-150, China) for 1 min at 20000 rpm, and the volume was adjusted to 30 mL with the addition of Trichloroacetic acid. The homogenate was filtered through 150 mm filter paper, and a specific volume reacted with the thiobarbituric acid reagent. HPLC was used to MDA quantification. Briefly, a total of 1 mL of extract and 3 mL of thiobarbituric acid reagent (40 mM dissolved in 2 M acetate buffer at pH 2.0) were mixed in a test tube and heated in a boiling water bath for 35 min. The reaction mixture was chilled prior to the addition of 1 mL of methanol, and 20 μL of the sample were injected into a Varian C18 HPLC column (5 μm, 150 × 4.6 mm) and held at 30 °C. The mobile phase consisting of 50 mM KH2PO4 buffer solution, methanol, and acetonitrile (72:17:11, v/v/v, pH 5.3) was pumped isocratically at 1 mL min-1.
All the revisions have been marked in yellow, green and blue (blue indicates the language revision). We are very grateful for your time and your favorable reconsideration.
Mr. You Zhou and Prof. Jiyun She
Central South University of Forestry and Technology
Emails: zhouyouhuhst@126.com; shejiyun2022@163.com

Reviewer 2 Report
- The synthesized spherical nano-TiO2 needs more characterization, and the description of SEM and XRD is not enough.
- Figure 1 c & d is not clear. Where is the characteristic diffraction peaks of TiO2? Author should mention it.
- There is no statistical analysis observed.
- Check the typo error.
Author Response
Dear Editor and Reviewer2:
Thank you for your letter and for the reviewers’ comments concerning our manuscript for publication in Nanomaterials. These comments are all valuable and very helpful for revising and improving our paper, as well as the important guiding significance to our researches. We have studied comments carefully and have made corrections point-to-point which we hope meet with approval and are marked with yellow, green and blue (blue indicates language revision). The responds to the reviewers’ comments are as follows:
We gratefully appreciate for the precious time the reviewer spent in making constructive remarks. Some more detailed explanation and discussion have been added into the revised manuscript and marked in yellow and blue (blue indicates language revision). Thank you so much for your useful comments. 1. The synthesized spherical nano-TiO2 needs more characterization, and the description of SEM and XRD is not enough. Author reply: Thanks for your advices. The same comment has been raised by reviewer 1#. Figure 1 has corrected to two separate Figures and the more explanations have been given in the revised manuscript and below (marked with yellow). Figure 1 SEM characterization of TiO2-bulk (a) and TiO2-rod (b).Figure 1 shows the SEM image of obtained TiO2 nanoparticles of spherical and rod morphology. The TiO2-bulk sphere has single shape and uniform dispersion. And the clear TiO2-bulk nanostructures can be seen having grain size of ~ 200 nm. Besides, it is clear show that TiO2-bulk consist of a number of crystallites which are seen by TEM image the synthesized TiO2 are spherical and rod-like, the size of nanoparticles is uniform, and the dispersion is good. XRD results are also the characteristic diffraction peaks of TiO2, Thus the SEM character-ization indicating indicates the successful synthesis of TiO2 with different morphologies [15].
Figure 2. The XRD patterns of TiO2-bulk (a) and TiO2-rod (b)
The XRD patterns of obtained TiO2 nanoparticles of spherical and rod morphology are shown in Figure 2(a) and (b) respectively. The synthesized nanoTiO2 both showed crystalline nature with 2θ peaks lying at 2θ=25.25o (101), 2θ=37.8o (004), 2θ=47.9o (200), 2θ=53.59o (105) and 2θ=62.36o (204). All the peaks in the XRD patterns can be indexed as anatase phases of TiO2 and the diffraction data were in good agreement with JCPDS files # 21-1272 [15]. So, the SEM and XRD characterizations illustrate the successful synthesis of desired TiO2-bulk and TiO2-rod nanomaterials. 2. Figure 1 c & d is not clear. Where is the characteristic diffraction peaks of TiO2? Author should mention it. Author reply: Thank for your valuable suggestion. The characteristic diffraction peaks of TiO2 have been added in the revised manuscript. And also can be seen in comment 2. Thank again for your advice. 3. There is no statistical analysis observed.Author reply: Thank for your valuable suggestion. In this study, the treated full seed were immersed in nano-TiO2 solution (TiO2-bulk and TiO2-rod) with diverse concentration (50, 100, 200 and 500 mg/L) at 45°C for 24hrs, pure water treatment was used as the control group. 30 seed with full particles and uniform size were selected from each group. The following determination of seed germination index and morphological index (Formulas of germination rate, G; germination potential, GP; germination index, GI and vigor index, VI), physiological indexes of seedling (Superoxide dismutase, SOD; peroxidase, POD; catalase, CAT and malondialdehyde, MDA content) and Fresh weight, root length, seedling height, are all based on the 30 seed derived seedings, the errors bar has been presented (as in Figure 3, Figure 4 and Table 1). 4. Check the typo error.Author reply: Thanks for pointing this out. The manuscript has been thoroughly revised and edited by a native speaker and typo errors have been carefully checked, so we hope it can meet the journal’s standard. Thank you so much for your useful comments.
All the revisions have been marked in yellow, green and blue (blue indicates the language revision). We are very grateful for your time and your favorable reconsideration.
Mr. You Zhou and Prof. Jiyun She
Central South University of Forestry and Technology
Emails: zhouyouhuhst@126.com; shejiyun2022@163.com

Round 2
Reviewer 1 Report
The manuscript is improved after the revision and it is now accaptable for publication.
Reviewer 2 Report
Author has satisfied all the queries. I recommend accepting the manuscript.